# Exercise Pills for Drug Addiction: Forced Moderate Endurance Exercise Inhibits Methamphetamine-Induced Hyperactivity through the Striatal Glutamatergic Signaling Pathway in Male Sprague Dawley Rats

**DOI:** 10.3390/ijms22158203

**Published:** 2021-07-30

**Authors:** Suryun Jung, Youjeong Kim, Mingyu Kim, Minjae Seo, Suji Kim, Seungju Kim, Sooyeun Lee

**Affiliations:** College of Pharmacy, Keimyung University, 1095 Dalgubeoldaero, Dalseo-gu, Daegu 42601, Korea; susu73@gw.kmu.ac.kr (S.J.); dbwjd1663@naver.com (Y.K.); cjjak009@naver.com (M.K.); seominjae1994@daum.net (M.S.); kimsuji921@naver.com (S.K.); tmdwn0502@naver.com (S.K.)

**Keywords:** methamphetamine, drug abuse, forced endurance exercise, hyperactivity, GSK-3β, glutamatergic signaling pathway

## Abstract

Physical exercise reduces the extent, duration, and frequency of drug use in drug addicts during the drug initiation phase, as well as during prolonged addiction, withdrawal, and recurrence. However, information about exercise-induced neurobiological changes is limited. This study aimed to investigate the effects of forced moderate endurance exercise training on methamphetamine (METH)-induced behavior and the associated neurobiological changes. Male Sprague Dawley rats were subjected to the administration of METH (1 mg/kg/day, i.p.) and/or forced moderate endurance exercise (treadmill running, 21 m/min, 60 min/day) for 2 weeks. Over the two weeks, endurance exercise training significantly reduced METH-induced hyperactivity. METH and/or exercise treatment increased striatal dopamine (DA) levels, decreased p(Thr308)-Akt expression, and increased p(Tyr216)-GSK-3β expression. However, the phosphorylation levels of Ser9-GSK-3β were significantly increased in the exercise group. METH administration significantly increased the expression of NMDAr1, CaMKK2, MAPKs, and PP1 in the striatum, and exercise treatment significantly decreased the expression of these molecules. Therefore, it is apparent that endurance exercise inhibited the METH-induced hyperactivity due to the decrease in GSK-3β activation by the regulation of the striatal glutamate signaling pathway.

## 1. Introduction

Methamphetamine (METH) is an amphetamine-type stimulant and is one of the most commonly abused drugs in the world [1]. METH causes short-term symptoms related to the sympathetic response of the autonomic nervous system (tachycardia, tachypnea, hypertension, dilatation of the pupils, hyperthermia, reduced fatigue), and increases euphoria, attention, energy, and libido while decreasing appetite [2,3,4,5]. However, repeated METH use can disrupt the central nervous system (CNS), causing neuropsychiatric changes, and in severe cases, death [6,7,8,9]. As drug abuse increases worldwide, deaths from METH misuse are also on the rise [10,11,12]. However, despite the increasing incidence of drug use disorders, there is no effective way to treat addiction [4]. To date, drug therapy, cognitive behavioral therapy, and social culture therapy have been used to treat drug addiction. However, these treatments have limitations. Drug use disorders such as methamphetamine addiction are characterized by consistent neurobiological changes at different stages of addiction [13]. In addition, recovery from addiction takes a long time, and recurrence is frequently observed. Current treatments cannot effectively be used to treat all the stages of drug addiction [14,15]. Therefore, a new approach is needed for drug addiction management.

Over the past few years, a number of studies have been performed on exercise-based interventions for METH abuse management [16]. Physical exercise has been reported to reduce the extent, duration, and frequency of drug use in drug-addicted individuals during the drug initiation phase, as well as during prolonged addiction, withdrawal, and recurrence [17]. Moderate aerobic exercise effectively reduced METH reuse by reducing depression and anxiety and controlled drug cravings in individuals addicted to METH [18,19,20]. In addition, exercise interventions had a positive effect during the METH withdrawal period [21,22,23,24]. Endurance and resistance exercise for eight weeks improved the subjects’ body composition and physical fitness level during the withdrawal period, and these physical changes reduced the drug dependence of the addicted individuals [25,26]. Similar results have been reported in animal studies. METH self-administration was significantly decreased in mice that were subjected to voluntary wheel running (1 h/day) [27], and the amount of METH self-administration also decreased in mice that performed voluntary wheel running prior to the self-administration session [28,29]. It would be important to have information on the neurobiological changes caused by exercise that may have a positive effect on drug-addicted individuals. According to previous studies, the neurobiological changes that take place in METH users through physical exercise may be due to the interaction of various changes, including the regulation of CNS neurochemicals [17,30], improvement of defense mechanisms involving antioxidants [31,32,33,34], neurogenesis [35], glial formation [36], and protection of the blood–brain barrier [37,38]. However, the results presented in these previous studies do not fully explain the effect of exercise on METH-induced neurobiological changes.

METH induces hyperactivity [39,40,41], a phenomenon called context-dependent behavioral sensitization, which plays a key role in the development of drug-seeking behavior [42,43,44]. METH-induced hyperactivity occurs due to the reinforcement of dopaminergic neurotransmission in the mesocorticolimbic system [45,46]. The mesocorticolimbic system consists of dopaminergic cell bodies in the ventral cortical region that project to the nucleus accumbens (NA, ventral striatum), dorsal striatum (caudate putamen), amygdala, hippocampus, and the prefrontal cortex. The neural adaptation process of the mesolimbic dopamine pathway occurs during repetitive drug exposure, affecting synaptic plasticity; this triggers neural adaptation of the neural circuits related to drug-induced learning and behavioral memory, and increases compulsive drug-seeking and recurrence [47,48]. A previous study suggested that glycogen synthase kinase-3 (GSK-3) in the striatum is a critical molecule in the acquisition of stimulant-induced activity and behavioral sensitization [49]. GSK-3 is a serine/threonine phosphorylating enzyme that is expressed throughout the mammalian brain and affects cell structure, motility, cell survival, apoptosis, neuronal differentiation, and DNA transcription [50,51]. It is highly regulated by phosphorylation at Tyr216 or Ser9 [52,53]. The phosphorylation of the Tyr216 residue occurs during the GSK-3β translation process, and results in the synthesis of the fully activated kinase; this activation process occurs in resting neurons to promote substrate access [54]. Ser9 phosphorylation is thought to be the main regulatory modification that occurs during the lifespan of the enzyme [55]. Ser9-phosphorylated GSK-3β remains inhibited, and dephosphorylation of the residue results in the activation of the kinase [56,57,58]. METH-induced GSK-3β activity is regulated by a number of signaling pathways (such as the dopamine, glutamate, and serotonin signaling pathways) [49]. Previous studies have shown that the dopamine receptor D2 (DRD2)–protein kinase B (Akt)–GSK3β signaling pathway is essential for the development of drug-induced behavioral sensitization [59,60,61,62]. Dopamine transporter knockout led to increased dopaminergic neurotransmission and hyperactivity in the knockout mice, which was accompanied by reduced Akt phosphorylation and reduced phosphorylation of the downstream substrates GSK-3α and GSK-3β; this further led to an increase in GSK-3β activity in the striatum [59]. Solis et al. [63] have reported that METH-induced locomotor sensitization is reduced in DRD2 knockout mice, and these mice have reduced locomotor response to a dopamine receptor D1 (DRD1) agonist. In addition, the glutamatergic receptor signaling pathway has also been reported to play an important role in conditioned place preference (CPP) induced by psychostimulants [64,65,66]. The activity of GSK-3β can be regulated by glutamatergic N-methyl-D-aspartate (NMDA) receptors. NMDA application to cultured hippocampal neurons produces a rapid dephosphorylate Ser9-GSK3β, demonstrating that NMDA receptor signaling activates GSK-3β [67].

Neuroadaptation to METH-induced behavioral sensitivity has been studied extensively, but there is little research into the effect of exercise training on METH-induced neuroadaptation. Therefore, this study investigated the effect of forced moderate endurance exercise on METH-induced behavior and the activity of GSK-3β via striatal dopaminergic and glutamatergic signaling pathways. As most individuals with drug addiction live a sedentary life, in this study, rats were subjected to forced exercise rather than voluntary exercise to mimic the conditions of exercise stress found in the clinic [68].

## 2. Materials and Methods

### 2.1. Animals

Male Sprague Dawley rats (Hyochang Science, Daegu, Korea) weighing approximately 200 g were housed in pairs. All rats were maintained under standard conditions with a constant room temperature (22 ± 1 °C), relative humidity (50 ± 10%), and a 12 h light/dark cycle (light from 07:00 to 19:00) with food (#2014, Harlan Telkad, Madison, USA) and water ad libitum. All rats were allowed to habituate to the novel environment for 1 week prior to the experiments. The rats were randomly assigned to four groups, as follows: saline control group (CON); methamphetamine injection group (MA); saline + exercise group (Ex); and methamphetamine + exercise group (Mex) (*n* = 8 in each group). The experiments were carried out in accordance with the guidelines approved by the Institutional Animal Care and Use Committee of Keimyung University (KM2020-008).

### 2.2. METH Administration and Exercise Training

#### 2.2.1. METH Administration

(+)-S-Methamphetamine hydrochloride (M8750, Sigma-Aldrich, Saint Louis, MO, USA) was dissolved in sterile 0.9% saline to a final concentration of 0.6 mg/mL. METH was freshly prepared before use. Rats in the MA and MEx group were administered 1 mg/kg METH–HCl intraperitoneally [69,70], and in the CON and Ex group, 0.9% saline was administered in the same volume. METH or saline was administrated once a day for 14 days. Every injection was performed between 09:00 and 12:00 p.m. The exercise group was administered METH or saline immediately after exercise training.

#### 2.2.2. Exercise Training

Exercise training was performed between 09:00 and 12:00 p.m. once a day for 2 weeks using an electric treadmill for laboratory animals (FT-200, Techman Soft, Taoyuan City, Taiwan). Exercise intensity was set by partially modifying the research methods of Wang et al. [71] and Kemi et al. [72]. Wang et al. [71] reported that the craving behavior for drugs in individuals addicted to METH was most effectively reduced with exercise of moderate intensity. Accordingly, the exercise training protocol consisted of moderate intensity (60% VO_2_max, 21 m/min, slope 0%) running on a treadmill for 60 min per day. During the first 7 days, the intensity and duration of the exercise were gradually increased to reach the target amount of exercise. After that, the exercise was performed while maintaining the target exercise amount until the end of the experiment. On the first day, exercise was performed on an electric treadmill at a speed of 10 m/min for 15 min to assist with adaptation to exercise. The duration and intensity of the exercise were then gradually increased by approximately 15% per day for 6 days until the target amount of exercise was reached. Since rats are more active in the dark, the front of the treadmill line was covered with dark paper. No electric shocks were used to reduce the stress effect of running on the treadmill during training sessions.

### 2.3. Experimental Design

All treatment and behavioral experiments were performed in the light cycle between 09:00 and 14:00. Body weight and food intake were measured daily during the 14-day experimental period. On the 14th day, the locomotor activity test was performed between 9:00 and 15:00. Exercise intervention was not performed on the day of the test, and METH or saline was injected immediately before locomotor activity test. On the 15th day, sampling commenced at 09:00. Before sampling, the Ex and MEx groups exercised lightly for 15 min at low intensity (10 m/min) to prevent loss of the exercise effect (Figure 1). Immediately after exercise, METH (1 mg/kg) or saline was injected and the sample was collected after resting for 2 h. CON and MA groups collected samples after 2 h of rest following saline or METH administration. For sampling, pentobarbital sodium (5 mg/100 g bwt) was administered intraperitoneally. Once rats were fully anesthetized, all blood was completely removed from the abdominal artery using a syringe, and the brain was then isolated by decapitation [73]. The right and left striatum were rapidly dissected out on ice. Tissues were immediately frozen in liquid nitrogen and stored at −80 °C.

### 2.4. Locomotor Activity

The locomotor activity test was performed from 9:00 on the 14th day of treatment, and the exercise group did not exercise on the day of the test. Rats were habituated to the testing boxes prior to testing by being handled in the colony room for 5 min daily for two days. They were then placed individually in a box for 30 min daily for another two days. On the day of the test, each rat was placed in a black Plexiglas square box (50 × 50 × 50 cm), and after an adaptation period of 15 min, 1 mg/kg of METH or saline was administered intraperitoneally. The locomotor activity was measured and displayed as total distance (cm) traveled during the 90 min period following administration of saline or METH. The apparatus was cleaned with 70% alcohol and dried after each session. The distance was measured by an automated video tracking system (Smart Video Tracking System- SMART 3.0, Panlab, Barcelona, Spain). The video files were analyzed by DigBehv analysis software to calculate motion tracking values, duration by movement speed, and total distance traveled. The speed of movement was divided into resting (<2.5 cm/s), slow (2.5–15 cm/s), and high (>15 cm/s). These classifications were guided by the results of previous studies [74,75].

### 2.5. Sample Preparation

The left and right striatum were homogenized together in an ice-cold RIPA buffer containing 250 mM sucrose, 10 mM HEPES/1 mM ethylenediaminetetraacetic acid (EDTA, pH 7.4), 1 mM Pefabloc (Roche), 1 mM EDTA, 1 mM NaF, 1 μg/mL aprotinin, 1 μg/mL leupeptin, 1 μg/mL pepstatin, 0.1 mM bpV (phen), 2 mg/mL glycerophosphate, protease inhibitors (#1861280, Fisher Scientific, Rockford, IL, USA), and a phosphatase inhibitor (P2850, Sigma-Aldrich, Saint Louis, MO, USA). Homogenization was performed on ice using a Wheaton tissue grinder (#357535 and 357537, DWK Life Sciences, Millville, NJ, USA). Homogenates were subjected to three freeze/thaw cycles and centrifuged for 15 min at 4 °C, 1000× *g*. The supernatant was separated and stored at −80 °C until assayed.

### 2.6. Western Blotting

Protein concentration was determined using the BSA assay (#5000006, Bio-Rad, Hercules, CA, USA). Aliquots were solubilized in Laemmli buffer (#1610747, Bio-Rad) and subjected to SDS/PAGE. A total of 40 ug of protein was separated by SDS-PAGE gel and immunoblotted with antibodies directed against GSK-3β (1:3000, Santa Cruz Bio, Dallas, TX, USA, sc-81462), phospho(Ser9)-GSK3β (1:1000, Santa Cruz Bio, sc-373800), phospho(Tyr279/216)-GSK3α/β (1:1000, Santa Cruz Bio, sc-81496), phospho(Ser473)-AKT (1:1000, Cell signaling, #4060), phospho(Thr308)-AKT (1:1000, Cell signaling, #13038), AKT (1:1000, Cell signaling, #2920), DRD1 (1:1000, Santa Cruz Bio, sc-33660), DRD2 (1:1000, Santa Cruz Bio, sc-5303), NMDA receptor 1 (NMDAr1, 1:1000, Santa Cruz Bio, sc-518053), calcium/calmodulin-dependent protein kinase kinase 2 (CaMKK2, 1:1000, Cell signaling, #16810), protein phosphatase 1 (PP1, 1:1000, Santa Cruz Bio, sc-7482), phospho(Thr180/Tyr182)-p38 mitogen-activated protein kinase (MAPK, 1:1000, Cell signaling, #4511), p38 MAPK (1:3000, Cell signaling, #9212), phospho(Thr202/Tyr204)-extracellular signal-regulated kinases 1/2 (Erk1/2, 1:1000, Cell signaling, #9101), Erk1/2 (1:3000, Cell signaling, #9102) and β-actin (1:1000, abcam, ab106814). After incubation with peroxidase-conjugated secondary antibody, blots were subjected to the enhanced chemiluminescent detection method with the ImageQuantTMLas 4000 imaging system (GE Healthcare). The relative intensity of the bands was assessed using SigmaGel (Jandel Scientific Corp., Schimmelbuschstrasse 25, Erkrath, Germany). The loading control was evaluated by the expression level of β-actin. The protein expression levels were normalized against β-actin protein levels and presented in arbitrary units (AU). Proteins activated by phosphorylation were presented as relative ratios of phosphorylation versus total expression level.

### 2.7. Striatal Dopamine Levels

Striatal dopamine levels were analyzed using a commercially available ELISA for dopamine according to the manufacturer’s instructions (Rat dopamine ELISA-kit, Fine Test, Wuhan Fine Biotech Co., Wuhan, China). The assay was performed in a 96-well plate, and absorbance was recorded using a plate reader (Spectramax). The dopamine levels were quantified and expressed as ng/mg tissue.

### 2.8. Statistical Analysis

Statistical analyses were conducted using GraphPad Prism 7.0 (GraphPad Software Inc., La Jolla, CA, USA). The results are presented as mean ± SE (standard error). Significant differences between groups were evaluated using one-way analysis of variance (ANOVA), and dietary intake and body weight were evaluated using two-way ANOVA. Post hoc validation was performed using Tukey’s test. A *p*-value < 0.05 was considered statistically significant. The analysis was performed by a statistician blinded to the experimental groups.

## 3. Results

### 3.1. Forced Moderate Endurance Exercise Significantly Reduced METH-Induced Hyperactivity

Dietary intake, body weight, and locomotor activity using an open field were measured to determine the effect of 14 days of exercise and METH administration on body composition and physical excitability. Endurance exercise or METH injection did not affect dietary intake and weight gain during the study period (Figure 2A,B). The administration of METH for 14 days significantly increased locomotor activity compared to the non-administration of METH (*p* < 0.001), while endurance exercise for 14 days did not affect locomotor activity (Figure 2C). In the MEx group that was administered combined treatment with endurance exercise and METH, the locomotor activity decreased significantly, though it was still significantly higher than that in the CON and Ex groups (*p* < 0.001, Figure 2C). When the locomotor activity was classified according to speed, the MA group moved exclusively at a fast speed (15 cm/s) for 60 min, whereas the CON and Ex groups moved 60% of the total travel time at a slow speed (2.5–15 cm/s). The animals took a break for the remaining 40% of the time (0–2.5 cm/s). The MEx group moved 5% of the total travel time at a fast speed, 90% at a slow speed, and took a break for the remaining 5% of the time (Figure 2D). In summary, 14 days of endurance exercise significantly reduced METH-induced hyperactivity in the MEx group. However, the activity level was not reduced to that in the CON group.

### 3.2. Forced Moderate Endurance Exercise Did Not Affect the METH-Induced Dopaminergic Signaling Pathway

Most of the drug-induced sensitization is considered to be regulated in the striatum that receive efferent processes from the dopaminergic mesocorticolimbic system [14,76]. The release of the neurotransmitters dopamine and glutamate in these core brain regions is essential for addiction-induced behavioral changes [77,78]. As per a previous study, we analyzed changes in striatal dopaminergic signaling molecules to understand the mechanism underlying the inhibition of METH-induced hyperactivity by endurance exercise training. METH administration for 14 days and/or endurance exercise significantly increased dopamine levels in the striatum (*p* < 0.01, Figure 3A). The expression levels of DRD1 and DRD2 were not significantly different between the four groups (Figure 3B,C). The expression level of p(Ser473)-Akt was not significantly different between the four groups (Figure 3D). However, the expression of p(Thr308)-Akt was significantly decreased in the MA, Ex, and MEx groups compared to CON (*p* < 0.05, Figure 3E). A significant increase in the expression of p(Tyr279)-GSK3α and p(Tyr216)-GSK-3β was observed in MA and MEx groups compared to CON (*p* < 0.01, 0.05, Figure 3F,G). The expression of p(Ser9)-GSK-3β, which is an inhibited from of GSK-3β, was not significantly altered in the MA group compared with the CON group, while it was significantly higher in the Ex and MEx groups than in the MA group (*p* < 0.05). The expression of p(Ser9)-GSK-3β in the Ex group was significantly higher than that in the CON group (*p* < 0.05, Figure 3H). To sum up the above results, 14-day METH and/or exercise treatment increased striatal dopamine levels, decreased the expression of p(Thr308)-Akt and increased the expression of p(Tyr206)-GSK-3β. However, the expression of p(Ser9)-GSK-3β, which inhibits the activity of GSK-3β, was significantly increased only in the exercise groups.

### 3.3. Forced Moderate Endurance Exercise Affected the METH-Induced Glutamatergic Signaling Pathway

To analyze the effect of endurance exercise on METH-induced hyperactivity via the glutamate signaling system, changes in the glutamate receptor and subsignal factors in the striatum were measured. According to previous studies, repeated administration of psychostimulants causes behavior sensitization, and the adaptation of dopaminergic and glutamatergic neurotransmission plays an important role in the expression of behavior sensitization [49,79,80]. In this study, two weeks of METH administration significantly increased the expression of NMDAr1 in the striatum compared to the other three groups (*p* < 0.01, 0.05, Figure 4A), and the expression levels of CaMKK2 and PP1, downstream of NMDAr1, were also significantly increased in the MA group (*p* < 0.05, Figure 4B,C). However, exercise intervention for two weeks significantly decreased the expression level of NMDAr1/CaMKK2/PP1 (*p* < 0.001, 0.01, 0.05, Figure 4A–C).

Previous studies have reported that Ca^2+^ enters cells through NMDA receptors by METH stimulation and activates MAPKs in the striatum [81]. Accordingly, we also analyzed the changes in the expression of MAPKs (p38, p42, p44). The level of p-p38MAPK expression was significantly increased in the MA group compared to the CON group, and decreased significantly in the Ex and MEx groups compared to the MA group (*p* < 0.05, 0.01, Figure 4D). The expression of p-ERK1 (p44MEPK) was not significantly different between the CON, MA, and MEx groups, but decreased significantly in the Ex group compared to the CON and MA groups (*p* < 0.05, Figure 4E). There was no significant difference in the expression of p-ERK2 (p42MEPK) between the MA and CON groups. However, p-ERK2 (p42MEPK) expression was significantly decreased in the Ex group compared to the CON and MA groups, and significantly decreased in the MEx group compared to the MA group (*p* < 0.01, 0.05, Figure 4F). In summary, two weeks of METH treatment increased the expression of NMDAr1/CaMKK2/PP1 and MAPKs, such as p-p38MAPK and p-ERK2, but endurance exercise training suppressed the expression of these molecules to the levels of the CON group.

## 4. Discussion

We investigated the effect of moderate endurance exercise (as used commonly in clinical studies) on METH-induced hyperactivity, and attempted to elucidate the underlying neurobiological mechanisms. METH administration for 14 days induced hyperactivity in SD rats, increased striatal dopamine levels, and increased GSK-3β activity by decreasing p(Thr308)-Akt expression and increasing p(Tyr216)-GSK3β expression levels. Interestingly, our results showed that the changes in the dopaminergic signaling pathway were similar between METH administration and METH administration combined with exercise. The combination of METH and endurance exercise increased the DA level in the striatum, decreased p(Thr308)-Akt, and increased the activity of GSK-3β by increasing p(Tyr216)-GSK3β. Previous studies have reported that physical exercise, similar to METH, can increase extracellular dopamine levels in the striatum and activate the neural circuits that underlie the reward and reinforcing effects [82,83]. Therefore, similar to the effect of chronic drug addiction, continuous physical exercise causes hedonic symptoms by activating the reward system [82,84]. In animal studies, the negative effects and symptoms of drug withdrawal were also observed in physical exercise withdrawal [85]. In mice exposed to wheel running, when access to the wheel was prohibited, systolic and diastolic blood pressure and body temperature decreased [86]; additionally, depression and behavioral symptoms such as anxiety were also observed [87,88,89]. However, the concept of “exercise addiction” is not yet clear and peer-reviewed evidence is lacking, so it has not been classified as a mental illness [90]. This study observed the effect of endurance exercise on METH-induced behavioral sensitization. Although the effect of exercise on METH-induced reward and reinforcement can be estimated, it is difficult to clarify the causal relationship.

In our study, the exercise group showed a significant increase in the expression of pSer9-GSK3β, unlike the METH treatment group. Thus, we may conclude that the inhibition of METH-induced hyperactivity caused by exercise is regulated by signaling pathways other than dopaminergic signaling pathways. Although the glutamate receptor is not a direct molecular target of METH, METH activates DRD1 to induce glutamate release [91,92], and the increase in extracellular glutamate activates the NMDA receptor; this in turn increases the intracellular influx of calcium [93,94]. The increase in intracellular calcium activates CaMK2, triggers calcineurin (PP2B), and subsequently dephosphorylates inhibitor-1 to induce PP1 activation [67]. PP1 directly increases GSK-3β activity by dephosphorylating Akt and Ser9-GSK3β [95,96,97,98]. In this study, METH administration for 14 days significantly increased the expression of NMDAr1 and PP1 in the striatum, and endurance exercise reduced these molecules to control (CON group) levels. Although DRD1 activity was not measured, we considered that METH-induced NMDAr1-PP1-GSK3β activity was inhibited by endurance exercise, and thus it may have acted as one of the regulatory factors of METH-induced hyperactivity. GSK-3β activation is regulated by a variety of factors, including the MAPK/ERK pathway [99]. METH administration for 14 days increased p38MAPK and ERK-2 phosphorylation in the striatum in this study, and endurance exercise inhibited METH-induced activation of the MAPKs. MAPK is a key signaling molecule in the regulation of pro-inflammatory cytokines and cellular responses to external stress. Rawas et al. [100] have reported that p38MAPK in the NA is involved in abnormal behavioral responses induced by drug abuse. Acute or chronic administration of psychostimulants such as cocaine or METH activates ERK throughout the striatum [101,102,103,104]. Gerdjikov et al. [105] suggest that ERK and p38 may be necessary for the establishment of NA amphetamine-produced CPP and may also mediate other forms of reward-related learning dependent on NA. It has been suggested that the activated MAPK cascade is involved in a major burst of gene expression that underlies long-term behavioral changes to drug exposure [106]. In this study, 14 days of METH administration increased the expression of NMDAr1, CaMKK2, and MAPK in the striatum, and endurance exercise suppressed the expression of these molecules. Therefore, these molecular changes may be involved in METH-induced hyperactivity. In summary, endurance exercise and METH had similar effects on the molecular expression of striatal dopaminergic markers, but opposite effects on glutamatergic markers. Although these molecular changes cannot fully explain the mechanism of the regulation of METH hyperactivity by exercise, it is possible to conclude that they can be implicated in this process.

The exercise intervention in this study involved forced moderate-intensity endurance exercise. In animal studies, voluntary exercise, in which the animal has access to freely running wheels, or forced exercise, in which the animal is made to run on a treadmill or swim in a closed pool, is employed. These methods are similar to human exercise patterns in a specific way, where voluntary exercise represents exercising according to an individual’s own will, and forced exercise represents standardized exercise formulated for the prevention or treatment of certain disorders [107]. According to a previous study, regular swimming exercise (45 min/day, 5 days/week, 14 days) decreased METH consumption in mice, and anxiety and depressive behavior decreased when METH administration was stopped [108]. In addition, CPP for amphetamine was inhibited after forced treadmill running [109], and both voluntary and forced wheel running were effective in reducing uncontrollable stress-induced learning deficits in mice [110]. There is very little research on the intensity of exercise therapy for the treatment of drug addiction. However, as most drug-addicted individuals have a sedentary lifestyle, low to moderate-intensity exercise is more likely to be preferred initially, rather than vigorous intensity exercise; moderate-intensity exercise has also been reported to be associated with higher compliance [111,112,113]. Based on these previous studies, we used moderate-intensity forced endurance exercise, and this exercise regimen effectively reduced METH-induced hyperactivity. However, it is difficult to formulate conclusions on the intensity and type of exercise that would be useful for drug addiction treatment based on the results of this study alone. Therefore, further research on the intensity, frequency, duration, and type of exercise is needed to formulate a therapeutic exercise regimen for individuals with METH addiction.

This study has a limitation in that it only targeted male rats. Numerous studies in humans [114,115,116,117,118,119] and rodents [120,121,122,123,124] have demonstrated gender differences in the effects of exercise on drug addiction. For example, female rats had reduced cocaine conditioned place preference (CPP) after chronic forced treadmill exercise, whereas male rats had complete inhibition of drug CPP after exercise [125]. In contrast, Ehringer et al. [120] reported a decrease in alcohol consumption after exercise in female rats, but not in male rats. In addition, other studies have shown a decrease in cocaine self-administration under extended-access conditions after exercise in both men and women [126,127]. The reasons for these conflicting results are not yet known, but may be related to the type of addictive drug, the type of motor activity [114,117], and gender-specific neurobiological mechanisms [123,128,129,130]. Overall, gender differences in the preventive and therapeutic effects of exercise on drug addiction are unclear and further investigations across both genders are needed.

## 5. Conclusions

Drug addiction is caused by a combination of psychological, physiological, and social factors, and an intervention combining various treatments will be effective. Therefore, we investigated the neurobiological effects of physical exercise on METH addiction to understand its therapeutic potential (“exercise pills”) for the treatment of drug addiction and prevention of recurrence. The results of this study show that forced moderate endurance exercise significantly reduces METH-induced hyperactivity in rats, which may be due to the regulation of the striatal glutamatergic signaling molecules leading to a decrease in GSK-3β activation (Figure 5). However, the results of this study alone cannot fully explain the mechanism of the effect of exercise on METH addiction. Future studies on the combined effect of various GSK-3β inhibitors (such as lithium, arylindolemaleimide, and thiadiazolidindiones) and exercise can help in the formulation of a combination therapy that can be used as an effective adjuvant treatment for drug addiction. In addition, using exercise as a therapeutic intervention requires additional research that can provide accurate guidelines (e.g., intensity, frequency, duration, type of exercise) for various types of drug-use disorders.

## Figures and Tables

**Figure 1 ijms-22-08203-f001:**
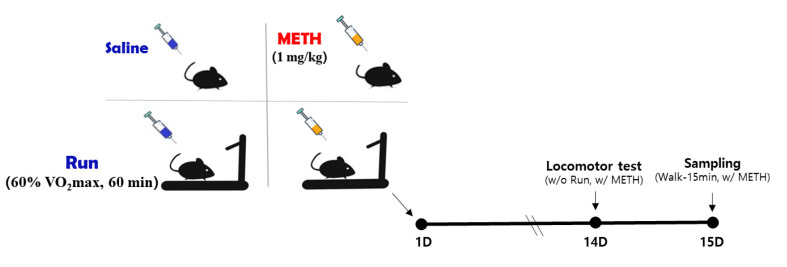
Schematic view of the experimental design. METH, methamphetamine; D, day; w/o, without; w/, with.

**Figure 2 ijms-22-08203-f002:**
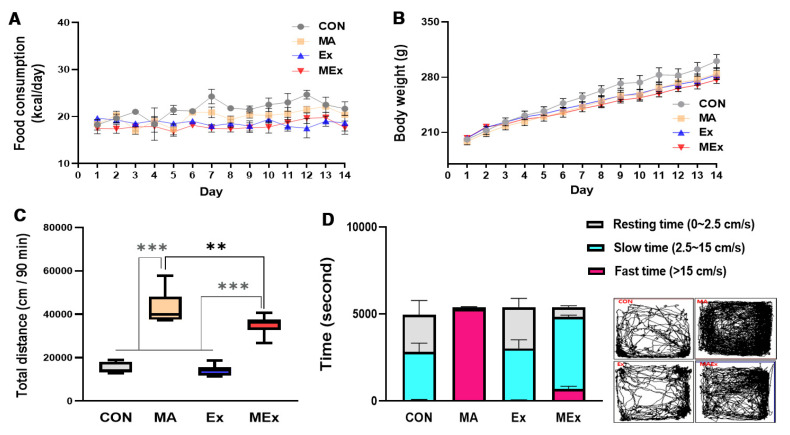
Forced moderate endurance exercise significantly reduced methamphetamine (METH)-induced hyperactivity. (**A**) Changes in food consumption (*n* = 8 per group, F_(39, 261)_ = 1.275). (**B**) Changes in body weight (*n* = 8 per group, F_(39, 248)_ = 1.744). (**C**) Total distance covered by the animal during the locomotor test conducted for 90 min (*n* = 7 per group, F_(3, 24)_ = 67.96). (**D**) The classification ratio according to the movement speed within the total travel time (*n* = 7 per group). Values are means ± SE, *** *p* < 0.001, ** *p* < 0.01. CON, saline control; METH, methamphetamine; MA, METH administration; Ex, saline + exercise; MEx, METH + exercise.

**Figure 3 ijms-22-08203-f003:**
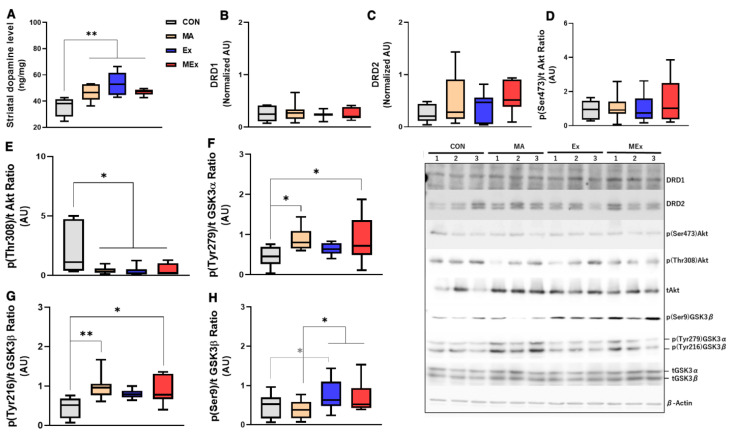
Forced moderate endurance exercise did not affect the methamphetamine (METH)-induced dopaminergic signaling pathway. Protein content was determined by Western blotting in the striatum of the brains of experimental rats. (**A**) Striatal dopamine level (*n* = 8 per group, F_(3, 28)_ = 10.25). (**B**) Dopamine receptor D1 (DRD1) expression level (*n* = 8 per group, F_(3, 28)_ = 0.2384). (**C**) Dopamine receptor D2 (DRD2) expression level (*n* = 8 per group, F_(3, 28)_ = 1.347). (**D**) p(Ser743)/t-Akt ratio (*n* = 8 per group, F_(3, 28)_ = 0.4629). (**E**) p(Thr308)/t-Akt ratio (*n* = 8 per group, F_(3, 28)_ = 6.311). (**F**) p(Tyr279)/t-GSK-3α ratio (*n* = 8 per group, F_(3, 28)_ = 3.433). (**G**) p(Tyr216)/t-GSK-3β ratio (*n* = 8 per group, (F_(3, 22)_ = 4.072), (**H**) p(Ser9)/t-GSK-3β ratio (*n* = 8 per group, F_(3, 28)_ = 3.241). Values are means ± SE, ** p* < 0.05, *** p* < 0.01. AU, arbitrary unit; CON, saline control; METH, methamphetamine; MA, METH administration; Ex, saline + exercise; GSK-3, glycogen synthase kinase-3; MEx, METH + exercise.

**Figure 4 ijms-22-08203-f004:**
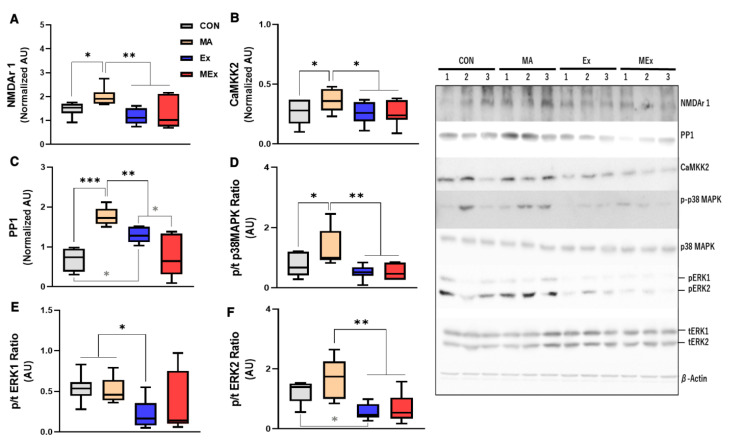
Forced endurance exercise training significantly reduced the methamphetamine (METH)-induced striatal glutamatergic signaling pathway. Protein content was determined by Western blotting in the striatum of brains of experimental rats. (**A**) NMDA receptor 1 (NMDAr1) expression level (*n* = 8 per group, F_(3, 27)_ = 6.718). (**B**) CaMKK2 expression level (*n* = 8 per group, F_(3, 26)_ = 2.489). (**C**) PP1 expression level (*n* = 8 per group, F_(3, 27)_ = 19.68). (**D**) p/t-p38MAPK ratio (*n* = 7 per group, F_(3, 24)_ = 7.042). (**E**) p/t ERK1 ratio (*n* = 7 per group, F_(3, 21)_ = 2.118). (**F**) p/t ERK2 ratio (*n* = 7 per group, F_(3, 23)_ = 5.471). Values are means ± SE, ** p* < 0.05, *** p* < 0.01, **** p* < 0.001. AU, arbitrary unit; CON, saline control; METH, methamphetamine; MA, METH administration; Ex, saline + exercise training; MEx, METH + exercise; NMDA, *n*-methyl-D-aspartate; CaMKK2, calcium/calmodulin-dependent protein kinase kinase 2; PPI, protein phosphatase 1; ERK, extracellular signal-regulated kinase; MAPK, mitogen-activated protein kinase.

**Figure 5 ijms-22-08203-f005:**
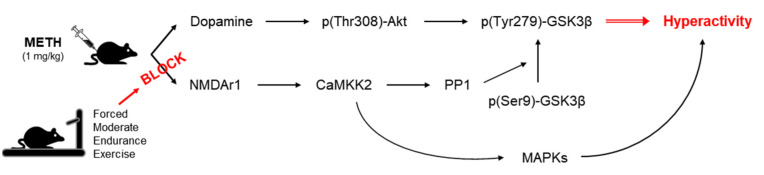
Summary of research results. METH, methamphetamine; NMDAr1, N-methyl-D-aspartate receptor 1; GSK, glycogen synthase kinase-3; MAPK, mitogen-activated protein kinase; CaMKK2, calcium/calmodulin-dependent protein kinase kinase 2; PPI, protein phosphatase 1.

## Data Availability

Available data are presented in the manuscript.

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
