# Peer review of "Exercise Pills for Drug Addiction: Forced Moderate Endurance Exercise Inhibits Methamphetamine-Induced Hyperactivity through the Striatal Glutamatergic Signaling Pathway in Male Sprague Dawley Rats"

_ijms, 2021, doi:10.3390/ijms22158203_

Round 1

Reviewer 1 Report

I think that the basic idea of the work is valid, in fact in many fields of medicine they are trying to use physical exercise in a therapeutic way

The work is well presented.

Were there any changes in behavior following exercise? In particular with regard to eating habits.

It has been rightly pointed out that the work would need to be continued, but one could hypothesize a workout that can be used for humans (duration, intensity ...) by staying on endurance or by considering other types such as HIIT or strength training. .

In this sense, a characterization, at least hypothetical, of the mechanism by which the exercise is effective would be interesting

Author Response

Thanks for the good evaluation.

Among the METH-administered group, the group that exercised had decreased excitability after administration compared to the METH group, and this phenomenon was verified through behavioral tests. And there was no significant change between groups in dietary intake.

We are conducting follow-up studies to find out the differences between various types of exercise, duration, and gender so that exercise can be used for the treatment and prevention of drug addiction. We will do our best to provide good follow-up research.

Reviewer 2 Report

The work presented by Jung and colleagues was nicely conducted, however, it does have few weaknesses that the authors should address before it can be accepted for publication.

  1. A limitation of the study is that the authors only used male rats and not also females, the possible effects of sex would have made this work unique and a critical addition to the existing literature. Thus, the authors should discuss and justify this limitation and should not use the oestrous cycle as an excuse. Divergent effects of sex on neural pathways activation/inhibition especially in addition are important and can open new roads to treatments and our understanding of such an important health issue.
  2. How will these findings correlate with those in animals exposed to Meth before exercise interventions?
  3. The abstract needs some improvements, as well as the text in general. For instance, the abstract contains too many methods details and some sentences read as fragments. This entire paragraph should be shortened: ‘A total of male Sprague–Dawley rats were randomly assigned to the following four groups: the control group, CON; methamphet-amine injection group, MA; endurance exercise group, Ex; and the methamphetamine plus endurance exercise group, MEx (N=10 in each group). The rats (except for the CON group) were treated with METH (1 mg/kg/day 19 METH-HCl, i.p.) and/or exercise (treadmill running, 21 m/min, 60 min/day) for 14 days. On the 14th day, a locomotor activity test was performed. The next day, the animals were decapitated under anesthesia with pentobarbital sodium (50 mg/kg), and the brains were collected. The striatum was separated from each brain, and the striatal tissue was analyzed by western blotting.’

The last part of the sentence reads incomplete, the pathways or proteins analysed should be mentioned.

The sentence “Forced moderate-endurance exercise sig- 29 nificantly reduced METH-induced hyperactivity in rats.” is repeated twice.

  1. The methods need additional details, the exercise protocol should be described in more detail.
  2. Did the exercise group receive saline injections? So that they will be habituated to the be injected and the injection on day 15th would not be stressfull? In the methods, there is no mention if they did and when (before or after exercising?), and for how long? two weeks? This group is described as “endurance exercise group, Ex” but later in the figure legends the same group (I believe) is named “Ex, saline+exercise training”.
  3. The authors should try to keep the description of the groups consistents.
  4. The authors report that the animals were exposed to moderate exercise (60% VO2max, 21 m/min, slope 0%) and that they were gradually habituated. Can the authors provide the initial values and the daily increases? Also was the habituation week part of the 14 days/2 weeks, or the animals were exposed to the “target exercise intensity” only for one week?
  5. When were the animals injected with saline or meth (time of day/night)?
  6. When was the locomotor activity measured (time of day/night)? Was the Meth/saline administered at the usual time before testing?
  7. The RIPA buffer composition should be described. After each freezing/thawing cycle were the pellets resuspended in the homogensation buffer, what detergent was used?Was the same sample preparation used for all the proteins of interest? The amount (ug) of proteins loaded should be reported in the methods. The authors need to describe how they controlled for equal loading. Did they use Ponceau staining? The normalization and determination of the expression levels should be described. Was the background subtracted from each band? Was the bands’ intensity for all the WB  normalized to beta-actin before calculating the ratios phosphorylated/total?  how were the changes in expression calculated, what does Normalised AU indicates for DRD1 and 2, NMDAr1 etc (3 and 4)?
  8. The antibody dilutions should be provided.
  9. The authors report using both the left and right striati for the Western Blots, what striati were used for the ELISA? Did the authors dived the animals and use half for each assay?
  10. There are different ANOVA statistical tests (one, two and three-way ANOVA), the authors should specify which one they used, likely One-way ANOVA.
  11. Were the behavioural observations performed by researched masked to the experimental group?
  12. The n value (number of animals/group) should be added to the figure legend for each assay/panel, Did all the animals survived? The authors show the bands from 3 animals/group, if this is the number of animals/group used, it is quite low. In the abstract, they report using 40 rats, 10animals/group, how many animals were used for the biochemical assays? How were the animals assigned?
  13. The use of Standard Deviation vs Standard Errors is now recommended when the sample number is low.
  14. The statistical test used to generate the P values should be added to the figure legend. Nowhere in the narrative or figure legends, the values generated by the statistical analysis (F, degree of freedom) are reported. These should be provided.
  15. The language should be thoroughly revised by a native speaker, for instance “in the dorsal and ventral striatal” should be “in the dorsal and ventral striatal REGIONS” or in “in the dorsal and ventral striatUM”.Sentences as “Protein content was determined by western blotting in the striatum of the brains of experimental rats.” are too wordy, there is no need to say “in the striatum of the brains” in the striatum should suffice.
  16. Abbreviations in the text and figure legends should be spelled out (m ; AU; ANOVA etc).
  17. Data in Figure 2D, how were the different times (resting slow fast) determined?
